# Influence of Profile Geometry on Frictional Energy Dissipation in a Dry, Compliant Steel-on-Steel Fretting Contact: Macroscopic Modeling and Experiment

Emanuel Willert

**Abstract:** Dry, frictional steel-on-steel contacts under small-scale oscillations are considered experimentally and theoretically. As indenting bodies, spheres, and truncated spheres are used to retrace the transition from smooth to sharp contact profile geometries. The experimental apparatus is built as a compliant setup, with the characteristic macroscopic values of stiffness being comparable to or smaller than the contact stiffness of the fretting contact. A hybrid macroscopic–contact model is formulated to predict the time development of the macroscopic contact quantities (forces and global relative surface displacements), which are measured in the experiments. The model is well able to predict the macroscopic behavior and, accordingly, the frictional hysteretic losses observed in the experiment. The change of the indenter profile from spherical to truncated spherical "pushes" the fretting contact towards the sliding regime if the nominal normal force and tangential displacement oscillation amplitude are kept constant. The transition of the hysteretic behavior, depending on the profile geometry from the perfectly spherical to the sharp flat-punch profile, occurs for the truncated spherical indenter within a small margin of the radius of its flat face. Already for a flat face radius which is roughly equal to the contact radius for the spherical case, the macroscopic hysteretic behavior cannot be distinguished from a flat punch contact with the same radius. The compliance of the apparatus (i.e., the macrosystem) can have a large influence on the energy dissipation and the fretting regime. Below a critical value for the stiffness, the fretting contact exhibits a sharp transition to the "sticking" regime. However, if the apparatus stiffness is large enough, the hysteretic behavior can be controlled by changing the profile geometry.

**Keywords:** elastic contact; friction; fretting; fretting regime; hysteresis; profile geometry

## 1. Introduction

Fretting is a widespread source of surface damage and, possibly, structural failure in many tribological contacts, which are subject to oscillating loads, e.g., in turbine blade roots [1] or artificial joints [2]. The physical (or chemical) nature and mechanisms of the tribological damage associated with fretting can be very diverse; in dry contacts of corrosion-resistant materials, the main sources of damage are (fretting) wear and (fretting) fatigue [3].

Based on so-called "fretting loops"—i.e., hysteresis loops between contact forces and macroscopic relative surface displacements—one can distinguish different fretting regimes: the partial slip regime with narrow hysteresis, the sliding regime with a broad (round or rectangular) hysteresis, and a mixed regime in between [4].

Although wear and fatigue in fretting contacts are separate (albeit interacting or even competing [5]) damage phenomena, in many cases, the partial slip regime is dominated by fatigue (due to the oscillating stress singularity at the edge of the permanent stick area), while the dominant phenomenon in the sliding regime is wear [6].

From a contact mechanical perspective, it is clear that the fretting regime (and, thus, the dominating damage mechanisms) can be heavily influenced by the contact profile, i.e.,

the geometrical form of the gap between the contacting surfaces in the undeformed state of first contact [7]. In that regard, the two main profile geometries that have been analyzed in the context of fretting are the parabolic (or spherical) (see, e.g., [8–10]) and the rounded flat punch profiles ([11,12])—which is due to the fact the these are the most common in engineering contacts that suffer from small oscillations.

Nonetheless, contact profile geometry as an influencing factor and, possibly, control variable in fretting has rarely been considered explicitly in the literature. Kim & Lee [13] theoretically studied the influence of the geometrical parameters in a plane rounded flat punch contact on the elastic edge crack propagation for the analysis of fretting fatigue failure. Bartha et al. [14] considered the influence of profile geometry on fatigue crack nucleation and propagation in an elastic fretting contact, validating their theoretical model with experimental results. Gallego et al. [15] systematically analyzed the wear-contact problem in fretting and evaluated the possibility of finding a corresponding optimal contact geometry. Warmuth et al. [16] experimentally studied the influence of the parabolic radius on the wear behavior in a steel-cylinder-on-steel-flat fretting contact. Zhang et al. [17] numerically compared the parabolic and rounded flat punch geometries in terms of the corresponding fretting behavior. Finally, Argatov & Chai [18] searched for the wear-optimal version of a symmetrical punch with compound curvature, which creates an almost constant contact pressure distribution in plane elastic contact.

Note that a related but slightly different question regards the influence of wear-induced profile changes on especially fretting fatigue. This problem, as an aspect of the complex interplay between fretting wear and fretting fatigue, has received a lot of attention in recent years (see, e.g., [19–21]).

Theoretically speaking, the most important distinction with respect to the macroscopic contact profile geometry, i.e., not considering roughness scales, is the one between (macroscopically) smooth profiles (e.g., paraboloids) and profiles with sharp edges (e.g., the flat punch). As smooth profiles exhibit significant partial slip (propagating from the contact boundary) but no stress singularities, they should be prone to wear rather than to fatigue, while sharp edges can inhibit slip at the cost of introducing stress singularities, which should make such profiles prone to fatigue rather than to wear.

To retrace the transition from smooth to sharp profile geometries—and, thus, to possibly overcome the wear-fatigue dilemma in fretting—very recently [22], the truncated parabolic geometry (which is also very easily realized experimentally) has been suggested as an interesting object of study in the context of fretting contacts. Changing the radius (or half-width) of the flat face of a truncated spherical (or cylindrical) profile geometry, one might be able to tune the transition from smooth to sharp contact gaps to combine "the best of both worlds".

Based on the fundamental scientific principle of separating different aspects or parts of a system or complex phenomenon, with the purpose of studying them individually, the "academic" analysis of fretting is usually concerned with laboratory contacts, which are not part of a larger macrosystem anymore (or the influence of the macrosystem is kept negligible). Accordingly, apparatuses for the experimental analysis of fretting contacts are usually constructed to be much stiffer than the contact itself, although it is known that system inertia and compliance can significantly influence tribological properties, e.g., wear rates [23]. While this approach is, of course, and undoubtedly, necessary for the understanding of the complex processes and mechanisms active in fretting contacts, it also discards the properties of the macrosystem as a possible control variable of the damage resulting from the tribological interaction. In contrast, for the analysis documented in the present manuscript, the experimental apparatus was built as a compliant setup, with the characteristic macroscopic values of stiffness being comparable to or smaller than the contact stiffness of the fretting contact that is to be studied during the analysis.

The aim of the analysis is, therefore, to study the influence of the profile geometry in fretting for a generic steel-on-steel oscillating laboratory contact, accounting for the mechanical properties (stiffness) of the macrosystem. Before studying local phenomena,

like wear or fatigue, the fretting regime shall be characterized based on the hysteretic (i.e., macroscopic) behavior, which is due to the frictional energy dissipation. A macroscopic model shall be built and validated, which can later be used as a component of a multiscale model for the analysis of the local phenomena.

The remainder of this manuscript is organized as follows: In Section 2, the experimental setup for the analysis is described in detail. Section 3 gives a hybrid macroscopic-contact-model that was used to predict the time development of the macroscopic contact quantities (forces and global relative surface displacements), which are measured in the experiments. In Section 4, the theoretical predictions are compared to experimental results to substantiate the model. After that, the model is used to analyze the influence of the profile geometry on the frictional energy dissipation, especially considering the high compliance of the experimental apparatus (which can have a significant qualitative impact, as will be demonstrated) in Section 5. A discussion of the methodology used and some conclusive remarks finish the manuscript.

## 2. Experimental Setup

A photograph of the experimental setup is shown in Figure 1. For a clearer understanding, a reduced scheme of the setup is given in Figure 2.

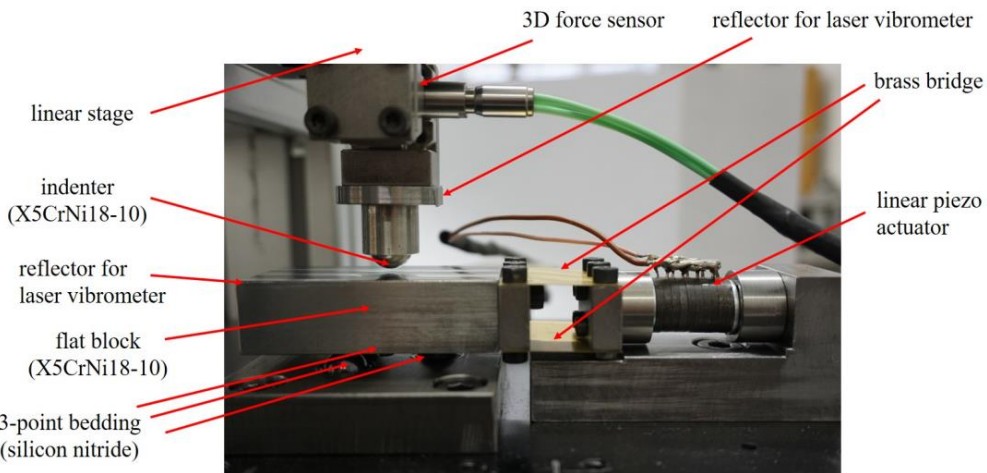

**Figure 1.** Photograph of the experimental setup.

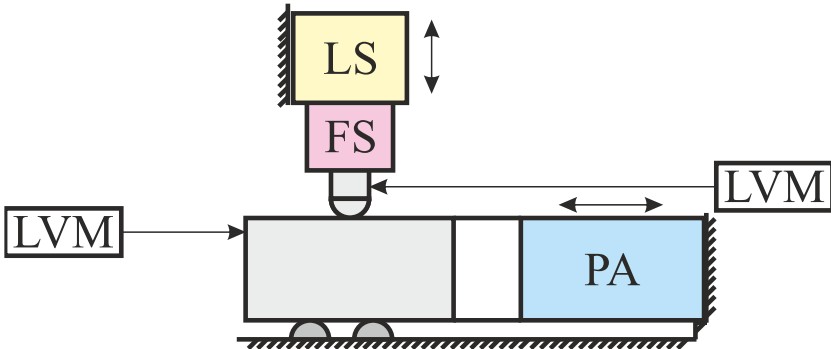

**Figure 2.** Scheme of the experimental setup with linear stage (LS), force sensor (FS), piezo actuator (PA), and laser vibrometers (LVMs).

A flat steel block (stainless steel X5CrNi18-10) rests on a three-point bedding of silicon nitride half-spheres, arranged as a regular triangle with the center of gravity below the center of gravity of the block. The point of first contact with the steel indenter (also stainless steel X5CrNi18-10, to ensure elastic similarity with the flat) lies above the center of gravity of the block. The normal axis of the apparatus consists of a 3D force sensor (Kistler 9317C, with

charge amplifier Kistler 5167A) and a linear stage (PI M-403.2DG, with controller PI C-863 Mercury). The force sensor is oriented in such a way that the sensor's normal axis—along which the sensor sensitivity is worse than along the sensor's tangential axes—coincides with the lateral axis of the contact so that the most relevant contact forces (in the contact's normal and tangential directions) are measured with the highest possible sensitivity. This setup also leads to a relatively high tangential compliance of this axis of the apparatus.

In the tangential direction (from left to right in Figures 1 and 2), the steel block is connected to a custom linear piezo actuator via a bridge of two thin brass plates—that is supposed to act as a parallel motion. The tangential displacements of the block and the indenter foundation are measured by two laser vibrometers (OFV-505 and OFV-503, with controllers OFV-5000). The data from the charge amplifier and the laser vibrometers are collected as analog input from a measuring board (NI USB-6353). The whole setup is controlled with LabVIEW scripts.

Before every experiment, the contact between the indenter and the flat steel block, as well as the contacts between the block and the silicon nitride half-spheres, are cleaned (the same bodies are used again for all experiments, as only a few dozen oscillations are performed in each experiment). Moreover, the contacts between the silicon nitride half-spheres and the flat block are lubricated with a drop of standard lubrication oil to reduce the corresponding coefficient of friction. Note that the research interest solely lies in the (dry) steel-on-steel contact between the indenter and the flat block.

For the experiments, two types of indenter profiles are used. First, the experiments are performed with full spheres. After that, the indenter profile is truncated, and a flat planform is introduced mechanically, in parallel to the contact plane.

The rms roughness of the original spherical steel indenter is approximately 1 μm, and the rms roughness of the flat steel block is approximately 3 μm. The rms roughness of the flat face of the truncated spherical indenter after the common turning process to generate the flat face is approximately 2 μm, so it is decided that no surface finish of the flat face is necessary. The tilting angle of the flat face with respect to the contact plane is less than $\pi/3600$. The width (in theradial direction) of the transition region from the flat part to the spherical part of the truncated indenter is less than the rms roughness of the surfaces.

## 3. Hybrid Macroscopic-Contact Model

As the energy dissipation in the fretting contact (as well as the fretting regime) can be determined from the hysteresis loops between contact forces and global displacements (i.e., macroscopic quantities), a hybrid macroscopic model was constructed that considers the equilibrium of the flat block and the indenter, and which is shown in Figure 3. As the frequency of the piezo actuator oscillation ($f$ = 20 Hz) is much smaller than the smallest eigenfrequencies of the block or the normal axis, inertial forces are neglected. In other words, the motion of the apparatus is considered in the quasi-static limit. Moreover, elastic deformations of the block as a whole are neglected. The brass plates are modeled as bent beams within the framework of the Euler–Bernoulli theory. All deformations are assumed to be small and within the elastic range.

The hybrid character of the model stems from the fact that all contact interactions (between the indenter and the block, as well as between the block and the silicon nitride bedding) are modeled within subroutines based on the method of dimensionality reduction (MDR) [24]. The MDR description of elastic tangential contact with friction operates within the framework of the Hertz–Mindlin-approximation ([25,26]), specifically:

- The characteristic length of the contact area is small compared to all linear dimensions of the contacting bodies; the gradients of the deformed surfaces in the vicinity of the contact are small (together, these two assumptions constitute the so-called "half-space-approximation").
- The frictional interaction can be described by a local-global Amontons–Coulomb friction with a constant coefficient of friction.

- All deformations are elastic. Elastic coupling between the normal and tangential contact problem and local lateral displacements are neglected.
- Effects of roughness or adhesion are neglected.

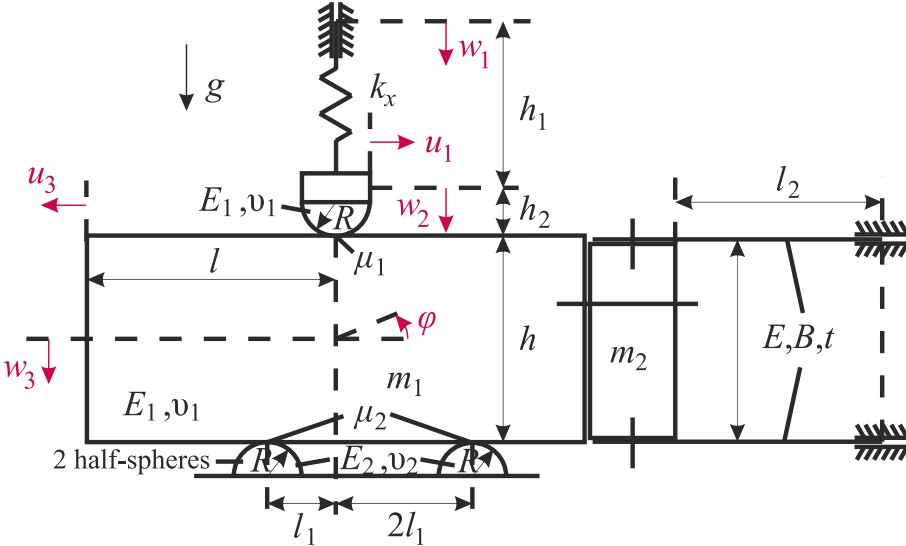

**Figure 3.** Macroscopic model of the experimental setup with relevant geometrical and mechanical quantities. The considered global displacements (i.e., degrees of freedom) are shown in red.

Within the MDR framework, the axisymmetric contact of elastic continua is exactly mapped onto the contact between a rigid plane profile with an elastic foundation of independent linear springs [27]. It has been proven that the MDR formalism, within the restrictions of the Hertz–Mindlin approximation, will provide the correct contact solution for arbitrary 2D oblique loading [28].

Within the model, the measured normal force (in the contact between the indenter and the flat) and the tangential displacement of the flat block (which is enforced by the piezo actuator) are taken as external excitations. For the determination of the fretting hysteresis loops, this leaves three degrees of freedom (the normal displacement $w_3$ of the block, the rotation $\varphi$ of the block, and the tangential displacement $u_1$ of the indenter) to be calculated from the vertical and angular equilibrium of the block, and the tangential equilibrium of the indenter. In that regard, Euler–Bernoulli theory will provide force laws for the vertical force and bending moment in the brass plates, and the MDR formalism provides (incremental) force laws for the contact interaction.

Hence, the macroscopic equation system is closed. Note that the real tangential indenter displacement in the contact is assumed to be $u_2 = u_1(h_1 + h_2)/h_1$ from the intercept theorem.

## 4. Comparison between Experimental and Numerical Results

As was stated above, the measured normal force in the contact between the indenter and flat, as well as the experimentally determined tangential displacement of the block, are taken as external loads/excitations for the model. That leaves the frictional force in the indenter–flat contact, as well as the tangential indenter displacement $u_1$, as variables that allow for a comparison between the model prediction and the experimental results.

The following parameters, shown in Table 1, are known a priori and were used for the model (the tangential stiffness $k_x$ of the normal axis of the apparatus was determined before the experiments):

As $b$ (i.e., the radius of the flat face of the truncated spherical indenter) is much larger than the characteristic contact radius for the full spherical (i.e., parabolic) profile under the considered nominal normal loads (below 1 N), the model results for the truncated sphere

can, actually, not be distinguished from the respective results, if a flat punch with the radius $b$ is used as the indenting body.

**Table 1.** List of a priori known parameters for the numerical simulation of the existing experimental apparatus (for notations, see Figure 3 and the text below).

| $l$ = 34 mm | $E_1$ = 200 GPa | $B$ = 42 mm | $h_1$ = 50 mm | $m_1$ = 335 g | $\nu_1$ = 0.29 |
|---|---|---|---|---|---|
| $l_1$ = 8 mm | $E_2$ = 300 GPa | $b$ = 0.6 mm | $h_2$ = 20 mm | $m_2$ = 47 g | $\nu_2$ = 0.29 |
| $l_2$ = 13 mm | $E$ = 100 GPa | $R$ = 6 mm | $h$ = 19 mm | $k_x$ = 1.9 N/μm | $t$ = 0.2 mm |

In Figure 4, a comparison between experimental results and model predictions for the fretting loops of frictional force over normal force (left) and frictional force over relative tangential displacement (right) are shown for a spherical indenter, with a nominal normal force of $F_{N,0}$ = 0.35 and a piezo oscillation amplitude of $u_A$ = 90 nm. The red line is the model prediction for a stationary fretting cycle with $\mu_1$ = 0.24 and $\mu_2$ = 0.06, and the black lines are the experimental results for seven fretting cycles in the initial stationary state (after 60 oscillation cycles to exclude the initial acceleration state, but before running-in). The choice of $\mu_1$ was based on the ratio of tangential force oscillation amplitude and nominal normal force (which varies about 0.01–0.02 between the three different experiments performed for the same parameter combination); $\mu_2$ was chosen arbitrarily but had only a very weak influence on the theoretical prediction for the hysteresis loop. Three experiments have been performed for each parameter combination. However, as the apparent coefficient of friction varies slightly between the experiments (even if they are done consecutively and with the same parameter combinations), the experimental hysteresis curves of only one experiment are shown to avoid confusion. The agreement between the model prediction and the experimental fretting loops is very good. The fretting contact is in the partial slip regime, with a narrow, slightly irregular hysteretic behavior.

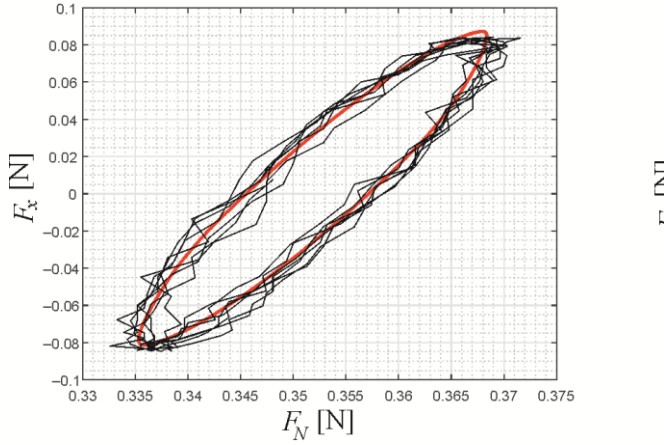 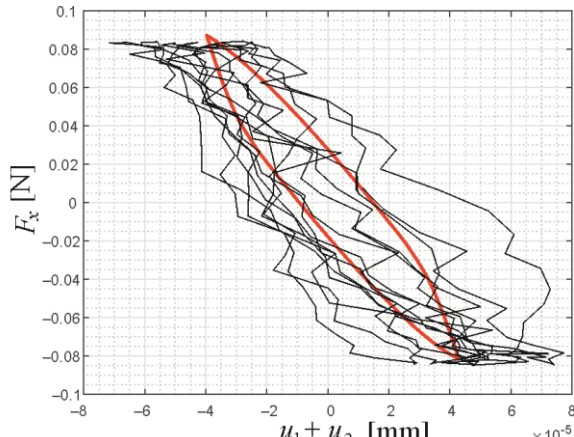

**Figure 4.** Comparison between experimental results and model predictions for the fretting loops of frictional force over normal force (**left**) and frictional force over relative tangential displacement (**right**) for a spherical indenter, with a nominal normal force of $F_{N,0}$ = 0.35 and an oscillation amplitude of $u_A$ = 90 nm. Red: model prediction for stationary fretting cycle with $\mu_1$ = 0.24 and $\mu_2$ = 0.06. Black: Experimental results for seven fretting cycles in the initial stationary state.

Figure 5 gives the comparison between experimental results and model predictions for the fretting loops for a spherical indenter, with a nominal normal force of $F_{N,0}$ = 0.61 and a piezo oscillation amplitude of $u_A$ = 220 nm. The model prediction uses $\mu_1$ = 0.26 and $\mu_2$ = 0.1. The agreement between theory and experiment is still very good, although slightly worse than in Figure 4. Note that the slight offset in the hysteresis curve (right) is not due to an offset of the displacement measurement (which, of course, is arbitrary) but

due to a misprediction of the phase difference between $F_x$ and $u_1$ (which is zero in the model—because the corresponding force law is simply the one of a linear spring—and nonzero in the experiments). However, the offset, naturally, has no influence on the prediction quality for the energy dissipation or the fretting regime.

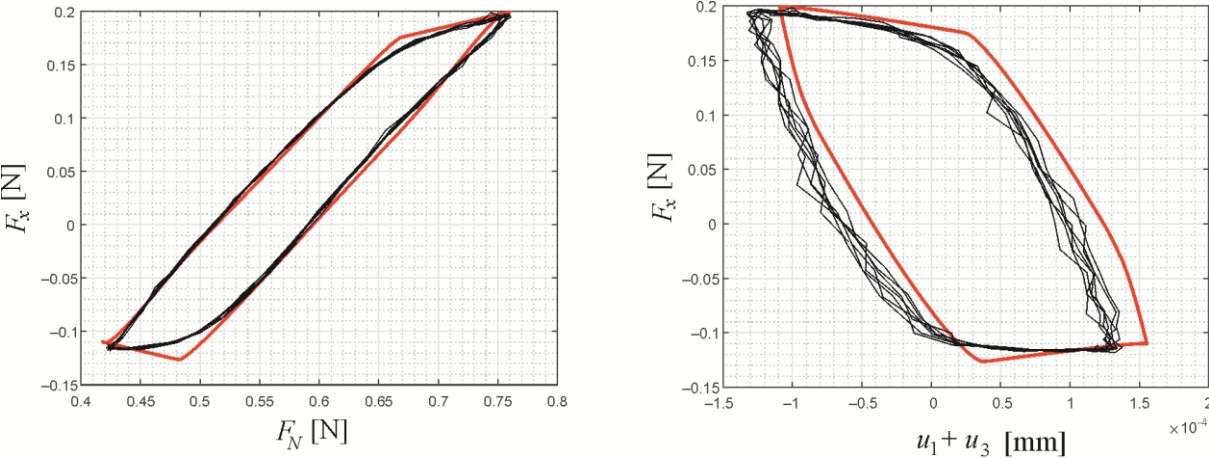

**Figure 5.** Comparison between experimental results and model predictions for the fretting loops of frictional force over normal force (**left**) and frictional force over relative tangential displacement (**right**) for a spherical indenter, with a nominal normal force of $F_{N,0} = 0.61$ and an oscillation amplitude of $u_A = 220$ nm. Red: model prediction for stationary fretting cycle with $\mu_1 = 0.26$ and $\mu_2 = 0.1$. Black: Experimental results for seven fretting cycles in the initial stationary state.

In Figure 6, a comparison is shown between experimental results and model predictions for the fretting loops of frictional force over normal force (left) and frictional force over relative tangential displacement (right) for a truncated spherical indenter, with a nominal normal force of $F_{N,0} = 0.31$ and a piezo oscillation amplitude of $u_A = 90$ nm. The model prediction uses $\mu_1 = 0.2$ and $\mu_2 = 0.1$. The agreement between theory and experiment is good; the fretting contact is in the gross slip regime, which is expected, as the contact stiffness is much higher, and the indentation depth $d$, therefore, much lower. Hence, the tangential displacement necessary to cause gross slip—which is of the order of $\mu d$—is also much lower.

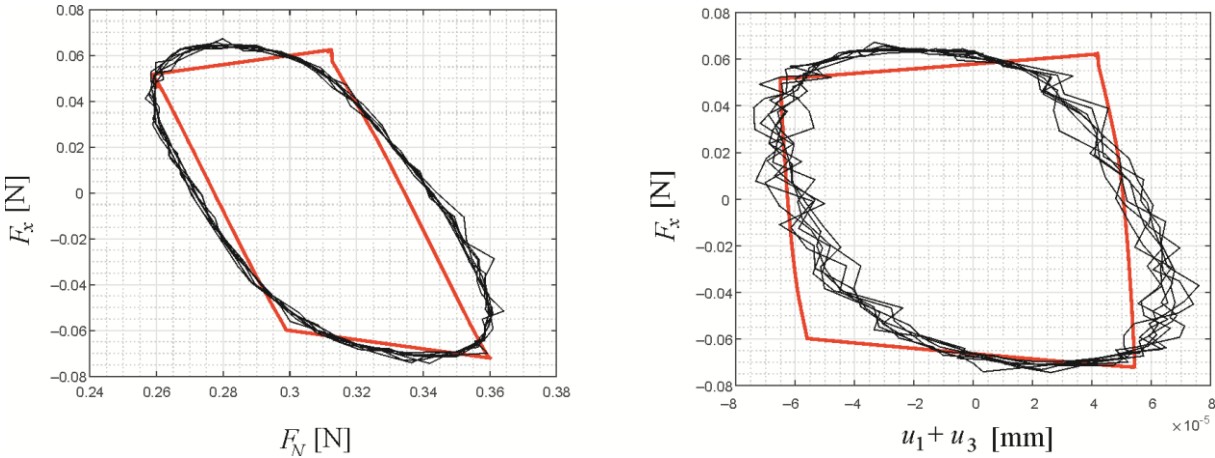

**Figure 6.** Comparison between experimental results and model predictions for the fretting loops of frictional force over normal force (**left**) and frictional force over relative tangential displacement (**right**) for a truncated spherical indenter, with a nominal normal force of $F_{N,0} = 0.31$ and a piezo oscillation amplitude of $u_A = 90$ nm. Red: model prediction for stationary fretting cycle with $\mu_1 = 0.2$ and $\mu_2 = 0.1$. Black: Experimental results for seven fretting cycles in the initial stationary state.

Figure 7 gives a comparison between experimental results and model predictions for the same fretting loops for a truncated spherical indenter, with a nominal normal force of $F_{N,0} = 0.61$ and a piezo oscillation amplitude of $u_A = 220$ nm. The model prediction uses the values $\mu_1 = 0.16$ (which is quite low) and $\mu_2 = 0.1$. The agreement is still good, and the contact is in the sliding regime. It can be noted that the behavior of the friction force at the "corners" of the hysteresis loop, i.e., at the beginning of the sliding phases, is much smoother in the experiments than predicted by the Amontons law.

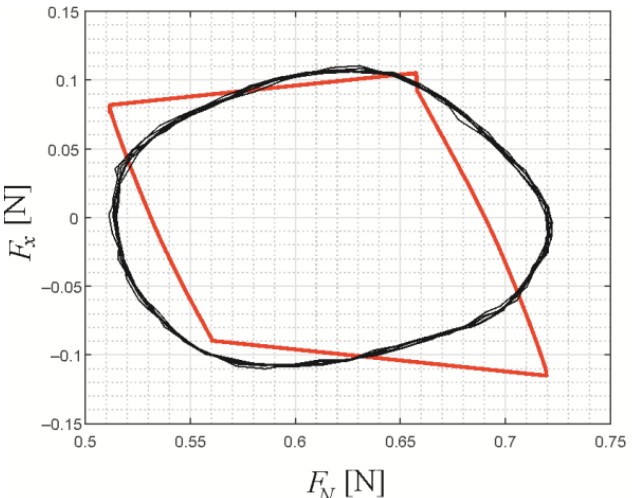
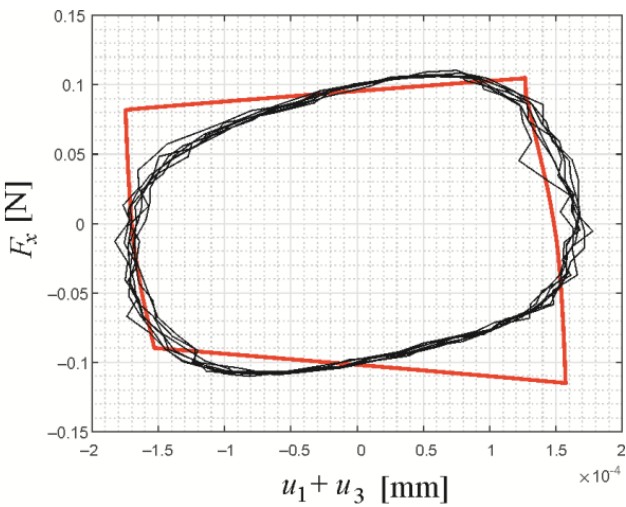

**Figure 7.** Comparison between experimental results and model predictions for the fretting loops of frictional force over normal force (**left**) and frictional force over relative tangential displacement (**right**) for a truncated spherical indenter, with a nominal normal force of $F_{N,0} = 0.61$ and a piezo oscillation amplitude of $u_A = 220$ nm. Red: model prediction for stationary fretting cycle with $\mu_1 = 0.16$ and $\mu_2 = 0.1$. Black: Experimental results for seven fretting cycles in the initial stationary state.

### 5. Analysis of Profile Influence Based on the Numerical Model

Let us analyze the system and the important influencing variables a little deeper, based on the macroscopic numerical model, which has been substantiated above.

The external excitations shall be harmonic, i.e., for the piezo-enforced tangential oscillation of the block,

$$u_3(t) = u_A \sin(2\pi f t), \tag{1}$$

and for the normal force,

$$F_N(t) = F_0 + \Delta F \sin(2\pi f t + \alpha). \tag{2}$$

Note that, in the experiments, the normal force oscillation is not externally controlled but exhibited by the apparatus due to the piezo actuator oscillation. However, $\Delta F$ and the phase angle $\alpha$ only have a weak influence on the frictional energy dissipation anyway.

To avoid showing the full fretting loops, let us introduce two quantities to characterize the frictional dissipation in the fretting contact between the steel indenter and the steel flat: the energy dissipation per cycle,

$$\Delta W = \left| \oint F_x \mathrm{d}(\Delta u) \right|, \tag{3}$$

with the frictional force $F_R$ and the macroscopic relative surface displacement $\Delta u$, as well as a slip index [29]

$$\delta = \frac{\Delta W}{4\mu_1 u_A F_0}. \tag{4}$$

A small slip index corresponds to the partial slip regime, while a slip index of approximately 1 corresponds to the gross sliding regime. Note that definition (4) slightly differs from the slip index proposed in the original paper by Varenberg et al. [29], as the dissipated energy is used directly, instead of slopes of the hysteresis curve, to account for the bimodal character of the fretting oscillation.

Out of the several dimensional variables, which will influence the frictional dissipation, $F_0$ defines the characteristic scales of the contact problem (indentation depth, contact radius, and, hence, contact stiffness) and shall therefore be kept constant. The normal force oscillation amplitude and phase angle, as well as the coefficient of friction between the steel flat and the silicon nitride bedding (if it is small enough), only have little influence on the dissipation in the fretting contact between the indenter and the flat. That leaves the actuator amplitude, $u_A$, the coefficient of friction between the indenter and the flat, $\mu_1$, the linear tangential stiffness of the normal axis of the apparatus, $k_x$, and the indenter profile as the main influencing properties. In that regard, the influence of $u_A$ is trivial (except for a very interesting stiffness effect, which will be discussed below): increasing the actuator amplitude will increase the energy dissipation and the slip index. Similarly, the influence of the friction coefficient $\mu_1$ is relatively simple, except for the aforementioned stiffness effect: increasing $\mu_1$ will generally reduce the slip index and (non-trivially) increase the energy dissipation. However, as for dry contacts of a given material pairing, the coefficient of friction can only be somewhat controlled within a very small margin; we shall not bother ourselves with the intricacies of these dependencies.

As was mentioned, there is an interesting stiffness effect, which, in fact, includes all important influencing factors, and which is due to the compliance of the experimental apparatus (or, generally speaking, the macrosystem). One can imagine the normal axis of the apparatus being very soft, i.e., $k_x$ being very small—but small compared to what? If

$$k_x u_A \ll \mu_1 F_0, \tag{5}$$

the axis is able to elastically follow the tangential excitation of the block without significant phases of sliding in the contact, and, therefore, the fretting contact will always be in the partial slip regime. In fact, the resulting slip index might suggest the existence of a "sticking regime" (without any frictional hysteresis), which, however, is only due to the apparatus compliance, as was pointed out before [30].

In other words, there is a critical apparatus stiffness of the order

$$k_c \approx \frac{\mu_1 F_0}{u_A}, \tag{6}$$

below which the indenter profile (via changing the contact stiffness) or other influencing variables cannot "pull" the fretting contact out of the partial slip (or "sticking") regime.

Let us illustrate these considerations with some small numerical parameter studies. Only $k_x$ and the indenter profile geometry were varied for the study; all other parameters and known quantities can be taken from Tables 1 and 2.

**Table 2.** List of chosen values for the fixed variables in the numerical parameter studies.

| $F_0 = 0.5$ N | $\Delta F = 0.1$ N | $\mu_1 = 0.2$ | $\mu_2 = 0.1$ | $u_A = 150$ nm |
|---|---|---|---|---|

As $F_0$, $\mu_1$, and $u_A$ are kept constant, the dissipated energy $\Delta W$ and the slip index $\delta$ incorporate the same information because the latter, in this case, is just a non-dimensional measure of the frictional hysteretic loss. Therefore, only the results of the macroscopic model for the slip index will be given.

In Figure 8, the numerical results for the slip index $\delta$ as a function of the radius $b$ of the flat face of the truncated spherical indenter (in logarithmic scaling) are shown for different values of the tangential apparatus stiffness. The thin lines of the respective

colors correspond to the numerical solution if a flat punch with radius $b$ is used as the indenting body. It can be seen that already for values $b > 30$ μm ($\log_{10}(30) \approx 1.5$) (which is approximately equal to the characteristic contact radius in the parabolic contact, under a normal load $F_0$), the solution for the hysteretic behavior cannot be distinguished from the respective solution for a flat punch indenter with radius $b$ of the flat face.

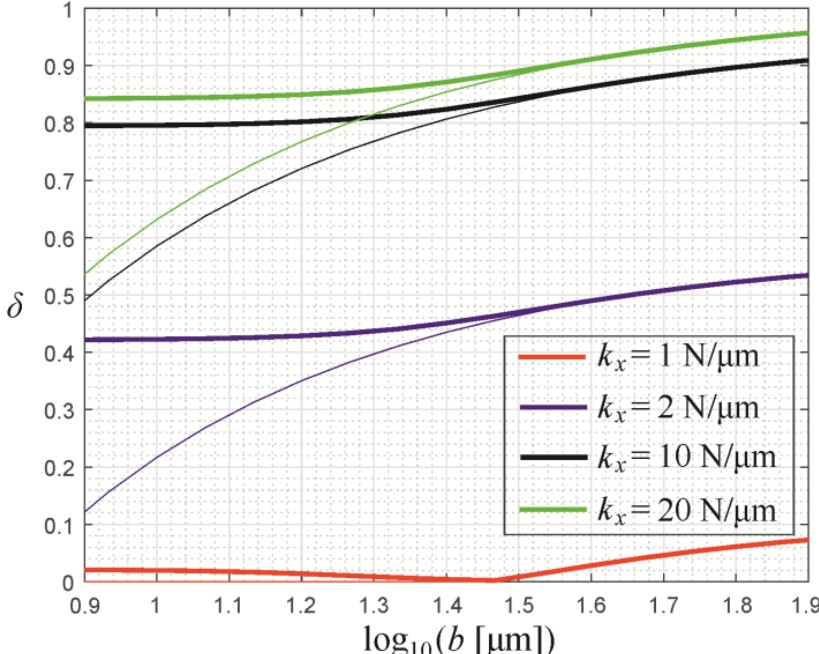

**Figure 8.** Slip index $\delta$ as a function of radius $b$ of the flat face of the truncated spherical indenter (in logarithmic scaling), for different values of the tangential apparatus stiffness $k_x$, according to the numerical model. The thin lines of the respective colors correspond to the numerical solution if a flat punch with radius $b$ is used as the indenting body.

Moreover, for $b < 10$ μm, the solution has already converged to the fully spherical (or parabolic) case, $b = 0$. In other words, the "window" for the geometrical transition in the hysteretic behavior from the smooth parabolic profile to a sharp flat-punch indenter is very small, independently of the apparatus stiffness. Changes in the hysteretic behavior for $b > 30$ μm are only due to the change in the contact stiffness of the flat-punch contact (which is proportional to its radius).

Also, it is apparent that—while the curves in Figure 8 for $k_x = 2$ N/μm, 10 N/μm and 20 N/μm are all qualitatively very similar—the curve for $k_x = 1$ N/μm is very different, and the slip index is close to (or even equal to) zero. This sharp transition is shown again in Figure 9, giving the slip index as a function of the tangential apparatus stiffness, for different values of the radius of the flat face of the truncated spherical indenter, according to the numerical model. For $k_x > 2$ N/μm, all three curves are in the mixed or sliding regime of the fretting contact, while between 1 N/μm and 2 N/μm, there is a steep transition to the "sticking" regime, where the dependence of the slip index on the apparatus stiffness somewhat resembles the behavior of the order parameter in a second-order phase transition. Notably, that resemblance is especially good in the case of the curves for $b = 50$ μm and $b = 100$ μm—for which the contact macroscopically already behaves like a flat-punch contact, as was discussed above—while the transition behavior for $b = 10$ μm (which basically corresponds to the spherical limit) is slightly smoother.

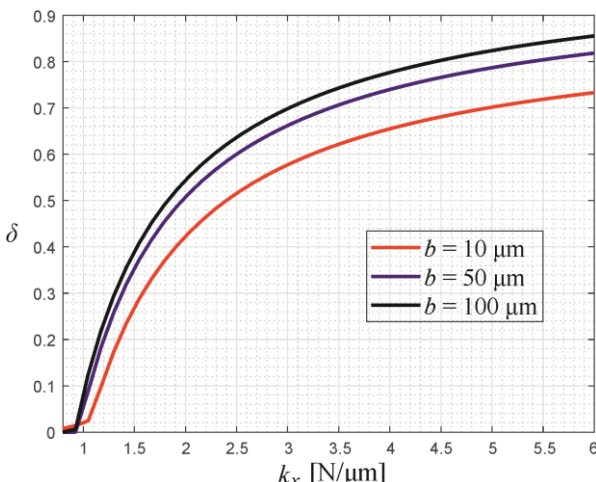

**Figure 9.** Slip index $\delta$ as a function of the tangential apparatus stiffness $k_x$, for different values of the radius $b$ of the flat face of the truncated spherical indenter, according to the numerical model.

## 6. Discussion and Conclusions

Being able to predict the macroscopic contact behavior in terms of forces, global relative displacements, and, as a result, energies is, of course, only the first step in the analysis of the fretting contact. The study of damage mechanisms inevitably requires knowledge of the behavior on (several) smaller scales and over much more oscillation cycles—in the present study, only the initial stationary states, before running-in, i.e., only several hundred cycles, were considered. It is, therefore, self-evident that the present study can only be the first step in a larger project, with the final aim of retracing the competition between wear and fatigue in steel-on-steel fretting contacts through the "lenses" of the contact profile geometry.

It is interesting that, while "fretting" is the result of many integrated, complex, tribological processes, a simple mechanical model with extremely few degrees of freedom, apparently, can successfully describe the contact's macro-behavior. This, at first glance, might be especially surprising, considering that the contact interaction is captured within the framework of a very basic contact mechanical description—the Hertz–Mindlin formalism. It must be noted, however, that the experimental setup was designed to comply with most of the restrictions of that formalism. For example, the normal load was kept very small to ensure the absence of relevant plastic deformations, equal materials were chosen for the contact pairing, and the contacts were cleaned before each experiment to avoid the influence of the tiniest amounts of wear debris in the contact.

On the other hand, the simplicity of the model also constitutes its strength, as all parameters of the model can be determined a priori, except for the global coefficients of friction (which are extremely difficult to predict precisely, anyway, for the type of contact considered here). In this regard, the model represents the "absolute minimum" of complexity—in the spirit of the principle to explain things "as simple as possible, but not any simpler"—which, in future work, can be extended step-by-step to improve its predictive power.

The next phase of research will be concerned with the influence of the profile geometry on the running-in process and the long-term wear behavior, studying the surface topography evolution and modeling the contact interaction on a smaller scale, based on the boundary element method—which has already been used to model adhesive wear within the framework of an asperity-free Rabinowicz criterion [31].

The main findings of the analysis documented in the present manuscript can be summarized as follows:

- The hybrid macroscopic–contact model is well able to predict the time behavior of the macroscopic contact quantities (forces and displacements) and, accordingly, the frictional hysteretic losses observed in the experiment.
- In the experiments and the simulation, the change of the indenter profile from spherical to truncated spherical "pushes" the fretting contact towards the sliding regime if the nominal normal force and tangential displacement oscillation amplitude are kept constant; this is mainly due to the higher contact stiffness (and, accordingly, smaller indentation depth) for the truncated profile.
- The transition of the hysteretic behavior, depending on the profile geometry from the perfectly spherical to the sharp flat-punch profile, occurs for the truncated spherical indenter within a small margin of the radius of its flat face. Already for a flat face radius which is roughly equal to the contact radius for the spherical case, the macroscopic hysteretic behavior cannot be distinguished from a flat punch contact with the same radius.
- The compliance of the apparatus (i.e., the macrosystem) can have a large influence on the energy dissipation and the fretting regime. Below a critical value for the stiffness, which is of the order of $\mu F_0/u_A$, the fretting contact exhibits a sharp transition to the "sticking" regime (with almost no dissipation). However, if the apparatus stiffness is large enough, the hysteretic behavior can be controlled by changing the profile geometry.

**Funding:** We acknowledge support by the German Research Foundation and the Open Access Publication Fund of TU Berlin. This research was funded by the German Research Foundation under project number PO 810/66-1.

**Institutional Review Board Statement:** Not applicable.

**Informed Consent Statement:** Not applicable.

**Data Availability Statement:** Not applicable.

**Acknowledgments:** The author is grateful to Valentin L. Popov for valuable discussions on the topic.

**Conflicts of Interest:** The author declares no conflict of interest.

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
