# Peer review of "Influence of Profile Geometry on Frictional Energy Dissipation in a Dry, Compliant Steel-on-Steel Fretting Contact: Macroscopic Modeling and Experiment"

_machines, doi:10.3390/machines11040484_

Round 1

Reviewer 1 Report

The overal concept, description of the experiment vs mathematical calculations are all sound. A very interesting paper, that may have practical usage in the mechanical design of fretting-sensitive contacts.  With this approach, it should be possible to also model other contacts, such as for instance electrical conductors, and determine the best combination of load, geometry, size for a given material combination.

Author Response

I am grateful for the kind comments of the reviewer. No concrete suggestions were made for the improvement of the manuscript.

Reviewer 2 Report

“Influence of Profile Geometry on Frictional Energy Dissipation in a Dry, Compliant Steel-on-Steel Fretting Contact: Macroscopic Modeling and Experiment” by E. Willert

The article is devoted to the experimental study of one of the tasks of contact mechanics associated with the impact of the indenter shape profile on the conditions and parameters of friction at the contact point. In this paper, interesting experimental data were obtained on measuring the forces at the contact point using indenters in the form of a sphere and truncated sphere. Also, a valuable practical conclusion was obtained, which consists in assessing the effect of the elastic compliance of the test system on the slip index and other friction parameters. The obtained quantitative values (see, figures 8 and 9) show a rather strong influence of apparatus stiffness on the friction parameters. This is an important conclusion, which shows not only the need to take into account the apparatus elastic compliance in precision measuring systems, but also the need to consider the changes in friction conditions caused by the value of this compliance.

In general, the article is well-structured, easy to read and understandable to the reader. There are no fundamental remarks on the merits of the paper. The only technical point of interest is how (using what processing technology) high surface quality and dimensions accuracy of the plane for the indenter in the form of a truncated sphere were ensured? What is the flatness tolerance for such indenter? It is known that the edge of the plane (the place of transition from the flat part to the spherical one) should be made accurately enough to minimize the influence of this edge on the results of the experiment.

The article leaves a positive impression and is recommended for publication in "Machines".

Author Response

I am grateful for the kind comments of the reviewer. All comments have been considered for the revised version of the manuscript. Detailed answers to the reviewer’s comments can be found below; changes in the manuscript have been highlighted in yellow.

Comment 2.1: The only technical point of interest is how (using what processing technology) high surface quality and dimensions accuracy of the plane for the indenter in the form of a truncated sphere were ensured? What is the flatness tolerance for such indenter? It is known that the edge of the plane (the place of transition from the flat part to the spherical one) should be made accurately enough to minimize the influence of this edge on the results of the experiment.

Answer: The rms roughness of the original sphere was approximately 1 μm, and the rms roughness of the flat block approximately 3 μm. The rms roughness of the flat face of the truncated spherical indenter after the common turning process to generate the flat face was approximately 2 μm, so it was decided that no surface finish of the flat face was necessary. The tilting angle of the flat face is less than π/3600. The width (in radial direction) of the transition region from the flat part to the spherical part of the truncated indenter is less than the rms roughness of the surfaces. So, considering that the theoretical model is macroscopic, the transition can be considered (macroscopically) perfect.

The information about the rms roughness values of the surfaces, as well as the geometrical properties of the truncated spherical indenter were added to the manuscript for the revised version.

Reviewer 3 Report

If the question is asked, then please try to answer (or/and to provide more information) in the journal paper.

1.       Abstract and the paper in general. Why the work was done? What application was simulated? “…the normal load was kept very small, to ensure the absence of relevant plastic deformations, equal materials were chosen for the contact pairing, and the contacts were cleaned before each experiment to avoid the influence of tiniest amounts of wear debris in the contact.”

2.       What was the roughness of the contacting bodies?

3.       Page 2 “apparatuses for the experimental analysis of oscillating contacts are usually constructed to be much stiffer than the contact itself.” Are you talking about dead-weight or servo-controlled type? What method of loading do you use? Inertia of the loading system can significantly influence the wear rate, see for example http://dx.doi.org/10.1016/j.triboint.2013.03.025

4.       Figure 2. If below are three spheres, then it is important to know where are two and where is one sphere in Figure 3.

5.       Figure 1. Please make it brighter since it is hard to see all details.

6.       Page 2. What contact is studied (steel-steel or steel-Si3N4)?

7.       Page 4 “two thin brass plates” Are they thick enough to provide movement in both directions?

8.       Page 4 “the contacts between the bedding and the flat block are lubricated with a drop of standard lubrication oil to reduce the corresponding coefficient of friction.” The paper is about dry sliding. Please explain what is “bedding”, “flat block”, “tangential direction”. It is possible to add some notes into Figure 3.

9.       Page 4 “the contact between the indenter and the flat, as well as the contacts between the flat and the bedding are cleaned” Have you used always “fresh” (new) bodies or you just cleaned and used the same bodies?

10.   Figure 1 and 3. In Figure 1 it is not clear that movement of “brass bridge” or “piezo actuator” is limited in vertical direction from upper side like it is shown in Figure 3.

11.   Table 1. May be it is possible to group (place them close to each other) the similar parameters like “l, l1, l2” and “B, b”.

12.   Page 6 and other places. What is “initial stationary state (i.e., before running-in)”? Are you talking about the first seven cycles (including initial acceleration during starting) or something else?

13.   Page 7. Have you measured COF (latin mu1 and mu2)?

14.   Figure 8, 9 and text. Please use same units when you refer to the Figure. Currently unitsa are different in text and in Figure (um, mm; N/um, N/mm)

Author Response

I am grateful for the kind comments of the reviewer. All comments have been considered for the revised version of the manuscript. Detailed answers to the reviewer’s comments can be found below; changes in the manuscript have been highlighted in yellow.

Comment 3.1: Abstract and the paper in general. Why the work was done? What application was simulated? “…the normal load was kept very small, to ensure the absence of relevant plastic deformations, equal materials were chosen for the contact pairing, and the contacts were cleaned before each experiment to avoid the influence of tiniest amounts of wear debris in the contact.”

Answer: The research was done without a specific application in mind; as fundamental research works on the influence of profile geometry in fretting are very scarce, a simple apparatus with a generic oscillating steel-on-steel contact was built to generally study the influence of contact profile geometry from a theoretical point of view. The restrictions/choices cited in the remark are purposefully introduced properties of the experimental setup, that were chosen to eliminate additional influences of, for example, plasticity or elastic coupling, but are not indented to simulate a concrete application system.

The following clarification has been added to the manuscript for the revised version in the “Introduction” section:

“The aim of the analysis is therefore to study the influence of the profile geometry in fretting, for a generic steel-on-steel oscillating laboratory contact, […]”

Comment 3.2: What was the roughness of the contacting bodies?

Answer: The rms roughness of the original sphere was approximately 1 μm, and the rms roughness of the flat block approximately 3 μm. The rms roughness of the flat face of the truncated spherical indenter after the common turning process to generate the flat face was approximately 2 μm, so it was decided that no surface finish of the flat face was necessary.

The information about the rms roughness values of the surfaces were added to the manuscript for the revised version.

Comment 3.3: Page 2 “apparatuses for the experimental analysis of oscillating contacts are usually constructed to be much stiffer than the contact itself.” Are you talking about dead-weight or servo-controlled type? What method of loading do you use? Inertia of the loading system can significantly influence the wear rate, see for example http://dx.doi.org/10.1016/j.triboint.2013.03.025

Answer: To my knowledge, in the context of fretting contacts, almost exclusively servo-controlled load application is used. In the manuscript’s experimental setup, the (nominal) normal load is applied via a servo-controlled linear stage, and the contact is excited by a piezo actuator, as described in the “Experimental Setup” section.

I am grateful to the reviewer for bringing that interesting reference on the influence of the macrosystem on tribological properties to my attention (although it stems from the context of sliding contacts).

The following clarification (with the corresponding reference) has been added to the manuscript for the revised version in the “Introduction” section:

“Accordingly, apparatuses for the experimental analysis of fretting contacts are usually constructed to be much stiffer than the contact itself, although it is known that system inertia and compliance can significantly influence tribological properties, e.g., wear rates [23].”

Comment 3.4: Figure 2. If below are three spheres, then it is important to know where are two and where is one sphere in Figure 3.

Answer: There are two spheres on the left side. Figure 3 was updated for the revised version.

Comment 3.5: Figure 1. Please make it brighter since it is hard to see all details.

Answer: The Figure has been updated for the revised version of the manuscript.

Comment 3.6: Page 2. What contact is studied (steel-steel or steel-Si3N4)?

Answer: Only the steel-on-steel contact is studied. The following clarification has been added to the manuscript for the revised version at the end of the “Experimental Setup” section:

“Note that the research interest solely lies on the (dry) steel-on-steel contact between the indenter and the flat block.”

Comment 3.7: Page 4 “two thin brass plates” Are they thick enough to provide movement in both directions?

Answer: They were chosen thin to mainly provide tangential movement. However, they have non-negligible bending stiffness.

Comment 3.8: Page 4 “the contacts between the bedding and the flat block are lubricated with a drop of standard lubrication oil to reduce the corresponding coefficient of friction.” The paper is about dry sliding. Please explain what is “bedding”, “flat block”, “tangential direction”. It is possible to add some notes into Figure 3.

Answer: Please see the answer to comment 3.6. Also, “bedding” was replaced by “silicon nitride half-spheres; “flat block” was replace by “flat steel block”, and the following clarification has been added to the manuscript for the revised version in the “Experimental Setup” section:

“In the tangential direction (from left to right in Figure 1 and Figure 2), […]”

Comment 3.9: Page 4 “the contact between the indenter and the flat, as well as the contacts between the flat and the bedding are cleaned” Have you used always “fresh” (new) bodies or you just cleaned and used the same bodies?

Answer: The same bodies were cleaned and used again. The following clarification has been added to the manuscript for the revised version in the “Experimental Setup” section:

“Before every experiment, the contact between the indenter and the flat steel block, as well as the contacts between the block and the silicon nitride half-spheres are cleaned (the same bodies are used again for all experiments, as only a few dozen oscillations are performed in each experiment).”

Comment 3.10: Figure 1 and 3. In Figure 1 it is not clear that movement of “brass bridge” or “piezo actuator” is limited in vertical direction from upper side like it is shown in Figure 3.

Answer: I agree with the reviewer. However, the actuator is fixed on the right-hand side, and has negligible bending compliance.

Comment 3.11: Table 1. Maybe it is possible to group (place them close to each other) the similar parameters like “l, l1, l2” and “B, b”.

Answer: The parameters have been slightly rearranged for the revised version of the manuscript.

Comment 3.12: Page 6 and other places. What is “initial stationary state (i.e., before running-in)”? Are you talking about the first seven cycles (including initial acceleration during starting) or something else?

Answer: The measurement starts after 3 seconds of oscillation, i.e. 60 cycles, to exclude the initial acceleration. The following clarification has been added to the manuscript for the revised version in the “Comparison […]” section:

 “[…] for seven fretting cycles in the initial stationary state (after 60 oscillation cycles, to exclude the initial acceleration state, but before running-in).”

Comment 3.13: Page 7. Have you measured COF (latin mu1 and mu2)?

Answer: No, not directly. The following clarification has been added to the manuscript for the revised version in the “Comparison […]” section:

“The choice of μ1 was based on the ratio of tangential force oscillation amplitude and nominal normal force (which varies about 0.01-0.02 between the three different experiments performed for the same parameter combination); μ2 was chosen arbitrarily, but has only very weak influence on the theoretical prediction for the hysteresis loop.”

Comment 3.14: Figure 8, 9 and text. Please use same units when you refer to the Figure. Currently units are different in text and in Figure (um, mm; N/um, N/mm)

Answer: Figures 8 and 9 have been updated for the revised version of the manuscript to comply with the notations in the text.

Reviewer 4 Report

An important point to clarify is the type of contact that has been studied. In the title of the paper, dry contact is mentioned, but later in the text it is stated that “a drop of standard lubrication oil” was used. In that case the type of contact does not appear to be dry as indicated in the title and should be changed.

Regarding the text on line 124 and in Figure 1, if the equipment configuration used is not the one shown in Figure 1, the image should be changed.

Regarding the experimental conditions, there are aspects that I consider important that are not sufficiently explained:

1) How many equal tests are performed?

2) Has the surface roughness of the steel block been measured, and could this value affect the results obtained?

3) Has the flat steel block been changed during the tests?

Line 290, the author indicates that the slip index expression is modified, but does not indicate how and why.

Line 337 and subsequent, the text speaks of contact radius dimensions in microns, when the graph uses mm and also in logarithmic scale. Please indicate the values of the graph in the text to help the reader understand what you want to explain.

Line 350, the units of kx in the text are N/micrometre while in figure 8 and 9 N/mm is used. Please use the same units in the text and in the figure for better understanding.

Author Response

I am grateful for the kind comments of the reviewer. All comments have been considered for the revised version of the manuscript. Detailed answers to the reviewer’s comments can be found below; changes in the manuscript have been highlighted in yellow.

Comment 4.1: An important point to clarify is the type of contact that has been studied. In the title of the paper, dry contact is mentioned, but later in the text it is stated that “a drop of standard lubrication oil” was used. In that case the type of contact does not appear to be dry as indicated in the title and should be changed.

Answer: Only the dry steel-on-steel contact is studied. The following clarification has been added to the manuscript for the revised version at the end of the “Experimental Setup” section:

“Note that the research interest solely lies on the (dry) steel-on-steel contact between the indenter and the flat block.”

Comment 4.2: Regarding the text on line 124 and in Figure 1, if the equipment configuration used is not the one shown in Figure 1, the image should be changed.

Answer: The Figure has been updated for the revised version of the manuscript.

Comment 4.3: How many equal tests are performed?

Answer: The following clarification has been added to the manuscript for the revised version in the “Comparison […]” section:

Three experiments have been performed for each parameter combination. However, as the apparent coefficient of friction is slightly varying between the experiments (even, if they are done consecutively and with the same parameter combinations), the experimental hysteresis curves of only one experiment are shown to avoid confusion.

Comment 4.4: Has the surface roughness of the steel block been measured, and could this value affect the results obtained?

Answer: The rms roughness of the flat block is approximately 3 μm. In the macroscopic model, surface roughness is not considered directly, but only via the resulting macroscopic coefficient of friction.

Comment 4.5: Has the flat steel block been changed during the tests?

Answer: No. The following clarification has been added to the manuscript for the revised version in the “Experimental Setup” section:

“[…] (the same bodies are used again for all experiments, as only a few dozen oscillations are performed in each experiment).”

Comment 4.6: Line 290, the author indicates that the slip index expression is modified, but does not indicate how and why.

Answer: The following clarification has been added to the manuscript for the revised version in the “Analysis […]” section:

“Note that the definition slightly differs from the slip index proposed in the original paper by Varenberg et al. [29], as the dissipated energy is used directly, instead of slopes of the hysteresis curve, to account for the bimodal character of the fretting oscillation.”

Comment 4.7: Line 337 and subsequent, the text speaks of contact radius dimensions in microns, when the graph uses mm and also in logarithmic scale. Please indicate the values of the graph in the text to help the reader understand what you want to explain.

Answer: Answer: The following clarification has been added to the manuscript for the revised version in the “Analysis […]” section:

“It can be seen that already for values b > 30 μm (log10(30) ≈ 1.5) […]”

Comment 4.8: Line 350, the units of kx in the text are N/micrometre while in figure 8 and 9 N/mm is used. Please use the same units in the text and in the figure for better understanding.

Answer: Figures 8 and 9 have been updated for the revised version of the manuscript to comply with the notations in the text.

Round 2

Reviewer 4 Report

The manuscript has been improved in this second version by including responses to the various comments made on the first version. Congratulations.